# Identification of new particle formation events with deep learning

Jorma Joutsensaari[1], Matthew Ozon[1], Tuomo Nieminen[1], Santtu Mikkonen[1], Timo Lähivaara[1], Stefano Decesari[2], M. Cristina Facchini[2], Ari Laaksonen[1,3] and Kari E.J. Lehtinen[1,4]

[1]Department of Applied Physics, University of Eastern Finland, P.O. Box 1627, FI-70211 Kuopio, Finland
[2]Institute of Atmospheric Sciences and Climate of the Italian National Research Council, Bologna, Italy
[3]Climate research Unit, Finnish Meteorological Institute, Helsinki, Finland
[4]Atmospheric Research Centre of Eastern Finland, Finnish Meteorological Institute, Kuopio, Finland

*Correspondence to*: Jorma Joutsensaari (jorma.joutsensaari@uef.fi)

**Abstract.** New particle formation (NPF) in the atmosphere is globally an important source of climate relevant aerosol particles. Occurrence of NPF events is typically analyzed manually by researchers from particle size distribution data day by day, which is time consuming and the classification of event types may be inconsistent. To get more reliable and consistent results, the NPF event analysis should be automatized. We have developed an automatic analysis method based on deep learning, a subarea of machine learning, for NPF event identification. To our knowledge, this is the first time when a deep learning method, i.e. transfer learning of a Convolutional Neural Networks (CNN), has successfully been used to classify automatically NPF events into different classes directly from particle size distribution images, similarly as the researchers do in the manual classification. The developed method is based on image analysis of particle size distributions using a pre-trained deep CNN, named AlexNet, which was transfer learned to recognize NPF event classes (six different types). In transfer learning, a partial set of particle size distribution images was used in the training stage of the CNN and the rest of images for testing the success of the training. The method was utilized for a 15-year long dataset measured at San Pietro Capofiume (SPC) in Italy. We studied the performance of the training with different training and testing image number ratios as well as with different regions of interest in the images. The results show that clear event (i.e., Classes 1 and 2) and non-event days can be identified with an accuracy of ca. 80 %, when the CNN classification is compared with that of an expert, which is a good first result for automatic NPF event analysis. In the event classification, the choice between different event classes is not an easy task even for trained researchers, thus overlapping or confusion between different classes occurs. Hence, we cross validated the learning results of CNN with the expert made classification. The results show that the overlapping occurs typically between the adjacent or similar type of classes, e.g., a manually classified Class 1 is categorized mainly into Classes 1 and 2 by CNN, indicating that the manual and CNN classifications are very consist for the most of the days. The classification would be more consistent, by both human and CNN, if only two different classes are used for event days instead of three classes. Thus, we recommend that in the future analysis, event days should be categorized into classes of "Quantifiable" (i.e. clear events, Classes 1 and 2) and "Non-Quantifiable" (i.e. weak events, Class 3). This would better describe the difference of those classes: both formation and growth rates can be determined for Quantifiable days but not both for Non-Quantifiable days. Furthermore, we investigated more deeply the days that are classified as clear events by experts and recognized as non-events by the CNN and vice versa. Clear

misclassifications seem to occur more commonly in manual analysis than in the CNN categorization, which is mostly due to the inconsistency in the human-made classification or errors in the booking of the event class. In general, the automatic CNN classifier has a better reliability and repeatability in NPF event classification than human-made classification and, thus, the transfer learned pre-trained CNNs are powerful tools to analyze long-term datasets. The developed NPF event classifier can be easily utilized to analyze any long-term datasets more accurately and consistently, which helps us to understand in detail aerosol-climate interactions and the long-term effects of climate change on NPF in the atmosphere. We encourage researchers to use the model in other sites. However, we suggest that the CNN should be transfer learned again for new site data with a minimum of ca. 150 figures per class to obtain good enough classification results, especially if the size distribution evolution differs from training data. In the future, we will utilize the method for data from other sites, develop it to analyze more parameters and evaluate how successfully CNN could be trained with synthetic NPF event data.

## 1 Introduction

Aerosol particles have various effects on air quality, human health and the global climate (Nel, 2005; WHO, 2013; IPCC, 2013). The air quality and health related problems are connected to each other, since in urban areas human exposure to elevated levels of particulate matter has been shown to cause respiratory problems and cardiovascular diseases (Brunekreef and Holgate, 2002), and eventually increase mortality (Samet et al., 2000). Very small particles like ultra-fine particles (smaller than 100 nm in diameter) can be particularly harmful because they can efficiently penetrate into the respiratory system and cause systemic effects (Nel, 2005). Air quality also affects visibility, for example, during smog episodes in large cities in Asia (Wang et al., 2013). On the global scale, aerosols affect the radiative balance of the Earth and therefore the climate. They affect the climate directly by either scattering incoming solar radiation back to space or by absorbing radiation. Indirectly, aerosols affect the climate via their role in cloud formation as cloud condensation nuclei (CCN). The number concentration and chemical properties of CCN particles affect both the brightness of clouds (Twomey, 1974) and their lifetime (Albrecht, 1989). Increased number of CCN are associated to smaller cloud droplets, which can lead to brighter and longer-lived clouds (Andreae and Rosenfeld, 2008). It has been estimated that both the direct and indirect aerosol climate effects cause a net cooling of the climate, and can therefore cancel out part of the global warming caused by greenhouse gases (IPCC, 2013).

Atmospheric new particle formation (NPF; formation and growth of secondary aerosol particles) is observed frequently in different environments in the planetary boundary layer (e.g. Kulmala et al., 2004). There are direct observations that NPF can increase the concentration of CCN particles regionally (Kerminen et al., 2012). Based on global aerosol model studies, it is estimated that 30–50% of global tropospheric CCN concentrations might be formed by atmospheric NPF (Spracklen et al., 2008; Merikanto et al., 2009; Yu and Luo, 2009). The longest continuous observational datasets of atmospheric NPF have been collected in Finland at the SMEAR stations in Hyytiälä and Värriö starting in 1996 and 1997, respectively (Kyrö et al., 2014; Nieminen et al., 2014), and at the GAW station in Pallas from 2000 onwards (Asmi et al., 2011). These three stations

are located in the Northern European boreal forest, and can be considered to represent rural and remote environments. In more anthropogenically influenced environments long-term NPF observations have been performed in Central Europe at Melpitz (Wang et al., 2017a) and in San Pietro Capofiume at the Po Valley basin in Northern Italy (Laaksonen et al., 2005; Hamed et al., 2007; Mikkonen et al., 2011). In high-altitude sites, which are at least part-time in the free troposphere, there are less and

shorter continuous measurements of NPF available. However, also in these high-altitude sites NPF has been observed to occur regularly (Kivekäs et al., 2009; Schmeissner et al., 2011; Herrmann et al., 2015). Recently, long-term NPF measurements have also been established in several new locations, e.g., in Beijing and Nanjing in China (Wang et al., 2017b; Kulmala et al., 2016; Qi et al., 2015) and in Korea (Kim et al., 2014; Kim et al., 2013).

Currently, all the NPF studies published in the literature have utilized visual-based methods to identify NPF events from the measurement data. Typically, these methods require 1–3 researchers to analyze periods of formation and growth of new modes in the size-distribution data. These methods were first introduced for analyzing data from the Finnish SMEAR stations by Dal Maso et al. (2005), and have been later slightly modified to suit analyzing data from different environments and measurement instruments (Hamed et al., 2007; Hirsikko et al., 2007; Vana et al., 2008). While the visual-based methods are in principle

simple and straightforward to apply, there are certain drawbacks in using them. First, they are very labor intensive, since the analysis of the aerosol size-distribution data is not automated. Second, these methods are somewhat subjective, i.e. different researchers might interpret the same datasets in slightly different ways. Finally, passing on the manual classification method from a researcher to a researcher could lead to an increasing systematic bias.

There have been attempts at improving NPF event classification methods and making them more automatic. In their comprehensive protocol article, Kulmala et al. (2012) introduced a concept for automatic detection of NPF events. This was based on identifying regions of interest (ROI) from the time-series of measured aerosol size-distribution data. These ROI were defined as time periods when elevated concentrations of sub-20 nm particles were observed compared to the concentration of larger particles. Developing the NPF classification to take into account also other data measured at the same site, such as

meteorological data and concentrations of trace gases, has allowed utilization of statistical methods to search for variables which could best explain and predict the occurrence of NPF. Hyvönen et al. (2005) and Mikkonen et al. (2006) applied discriminant analysis for multi-year datasets of aerosol size-distributions and several gas and meteorological parameters measured at Hyytiälä, Finland and San Pietro Capofiume, Italy, respectively. Both of these studies were able to find the characteristic conditions for NPF event days in each site and it was seen that the conditions differ significantly. They were also

able to construct models to predict the probability of NPF occurrence with reasonable accuracy, and this approach has also been used in the day-to-day planning of a complex airborne measurement campaign (Nieminen et al., 2015). Junninen et al. (2007) introduced an automatic algorithm based on self-organizing maps (SOM) and a decision tree to classify aerosol size distributions. More recently, preliminary attempt at utilizing machine learning on big datasets have been reported (Zaidan et al., 2017).

There exists a long list of algorithms to automate the classification of different type of datasets (Duda et al., 2012), such as k-means, Support Vector Machine (SVM), Boltzmann Machine (BM), decision trees, etc. Our method choice is the deep feedforward Neural Network (NN). The idea of NN is not new, as the idea was first brought to life by Hebb in 1949 (Hebb,

2005). Since then, this field has drawn the attention of many researchers (Farley and Clark, 1954; Widrow and Hoff, 1960; Schmidhuber, 2015) probably because of its apparent simplicity and versatility. In this study, we used a convolutional NN (CNN) because it mimics the visual cognition process of humans. One of the main bottlenecks in the use of the NNs is the learning stage of NN; NN must be trained before it can be used in image recognition or other applications. For instance, thousands of images are typically needed for learning in image recognition applications. To overcome this problem, we used

a pre-trained deep CNN, named AlexNet, which was originally trained to recognize different common items (e.g., pencils, cars and different animals) and we transfer learned it to recognize images of different NPF event types. The transfer learning reduces significantly the number of images needed in the training process, from thousands to hundreds. The CNN used in this study and transfer learning of the CNN are described in detail in the Appendix A and Fig. 2.

In atmospheric science, several studies have used deep learning or other novel machine learning methods in data analysis. Deep learning and other machine learning algorithms are commonly used in remote sensing (Zhang et al., 2016; Lary et al., 2016; Han et al., 2017; Hu et al., 2015). Remote sensing is very suitable for machine learning because large datasets are available and the theoretical knowledge is incomplete (Lary et al., 2016). For instance, Han et al. (2017) introduced a modified pre-trained AlexNet CNN and Hu et al. (2015) used several CNNs (e.g. AlexNet and VGGnets) for remote sensing image

classification. Ma et al. (2015) used transfer learning in a Support Vector Machine (SVM) approach for classification of dust and clouds from satellite data. Other applications in atmospheric science include, e.g., air quality predictions (Li et al., 2016; Ong et al., 2016), characterization of aerosol particles with an ultra violet-laser induced fluorescence (UV-LIF) spectrometer (Ruske et al., 2017) and aerosol retrievals from ground-based observations (Di Noia et al., 2015; Di Noia et al., 2017). In addition, we recently applied different machine learning methods (e.g. neural network and SVM) for aerosol optical depth

(AOD) retrieval from sun photometer data (Huttunen et al., 2016). Although machine and deep learning approaches have already been used in several applications in atmospheric science, the use of those novel artificial intelligence methods will expand rapidly in the future. Nowadays, those methods are more efficient and easier to use due to the development of user-friendly applications and increasing computing capacity (e.g. Graphics Processing Units, GPUs), and they can be applied in problems that are more complicated.

Here, we demonstrate that a novel deep learning-based method, i.e. transfer learning of a commonly used pre-trained CNN model (AlexNet), can be efficiently and accurately used in classifying of new particle formation (NPF) events. The method is based on image recognition of daily-measured particle size distribution data. To our knowledge, this is the first time when a deep learning method, i.e. transfer learning of a deep CNN, has been successfully used in an automatic NPF event analysis for

a long-term dataset. We will show that the deep learning-based method will increase the quality and reproducibility of event analysis compared to manual, human-made visual classification.

## 2 Materials and methods

### 2.1 Measurement site, instrumentation and dataset

In this study, we analyzed a long-term particle size distribution (PSD) dataset measured at the San Pietro Capofiume (SPC) measurement station (44° 39' N, 11° 37' E, 11 m a.s.l.), Italy. The PSD measurements started on 24 March 2002 at SPC and have been uninterrupted, except for occasional system malfunctions. The SPC station is located in a rural area in Po Valley about 30 km northeast from the city of Bologna. The Po Valley area is the largest industrial, trading and agricultural area in Italy with a high population density and hence it is one of the most polluted area in Europe. On the average at SPC, NPF events
occur on 36 % of the days whilst 33 % are clearly non-event days and the probability for NPF events is highest in spring and summer seasons (Hamed et al., 2007; Nieminen et al., 2018).

At SPC, PSDs are measured with a twin Differential Mobility Particle Sizer (DMPS) system; the first DMPS measures PSDs between 3 and 20 nm and the second one between 15 and 600 nm. The first DMPS consists of a 10.9 cm long Hauke-type
differential mobility analyzer (DMA)(Winklmayr et al., 1991) and an ultrafine condensation particle counter (CPC, TSI model 3025) whereas the second DMPS consists of a 28 cm long Hauke-type DMA and a standard CPC (TSI model 3010). One measurement cycle lasts for 10 min. The PSDs used in this study were calculated from the measured data using a Tichonov regularization method with a smoothness constraint (Voutilainen et al., 2001). The data from two different DMPS instruments was combined in the data inversion. The CPC counting efficiency and diffusional particle losses in the tubing and DMA
systems were taken into account in the data analysis. In addition to PSD measurements, several gas and meteorological parameters are continuously measured at the SPC station (e.g. $SO_2$, NO, $NO_2$, $NO_x$, $O_3$, temperature, relative humidity, wind direction, wind speed, global radiation, precipitation, and atmospheric pressure). The measurement site and instrumentation have been described in detail in previous studies (Laaksonen et al., 2005; Hamed et al., 2007).

The analyzed dataset covers 4177 days (files) from the start of the measurement in SPC on 24 March 2002 until 16 May 2017 (totally 5534 days). The number of days at the different NPF classes and division into training and testing categories are summarized in Table 1.

### 2.2 Classification of new particle formation events (traditional method)

Currently, a classification of NPF events is practically made manually, i.e., researchers visually inspect contour plots of time
series of aerosol size distribution data and time evolution of nucleation-mode particles (particle dimeter below ca. 50 nm) (Kulmala et al., 2012). For the dataset of this study, the manual classification of NPF events into different categories is based

on guidelines described by Mäkelä et al. (2000) and Hamed et al. (2007). Figure 1 shows examples of measured time series of PSDs (time in x-axis, particle diameter in y-axis and particle concentration presented by different colors) for different event classes.

5 In the first step of event analysis, data is classified into days with NPF events and days without particle formation (non-event days, NE). A day is considered as a NPF event day if formation of new aerosol particles starts in the nucleation mode size range (<25 nm) and subsequently grows, and the formation and growth is observed for several hours. The NPF event days are further classified according to the clarity and intensity of the events (Hamed et al., 2007):

- **Class 1 events** (Class_01) are characterized by high concentrations of 3–6 nm particles with only small fluctuations
10 of the size distribution and no or little pre-existing particles in the smallest size ranges. Class 1 events show an intensive and clear formation of small particles with continuous growth of particles for seven to ten hours.

- **Class 2 events** (Class_02) show the same behavior as Class 1 but with less clarity. The formation of new particles and their subsequent growth to larger particle sizes can be clearly observed but, e.g., fluctuations in the size distribution are larger. Furthermore, the growth lasts for a shorter time than for Class 1 being about five hours on
15 average. In event Classes 1 and 2, it is easy to follow the trend of the nucleation mode and, hence, the formation and growth rates of the formed particles can be determined confidently.

- **Class 3 events** (Class_03) include days when same evidence of new particle formation can be observed but growth is not clearly observed. For example, the formation of new particles and their growth to larger particle sizes may occur for a short time but is then interrupted (e.g. by a drop in the intensity of solar radiation, rain). In addition, the
20 days with weak growth are classified in that category

The classification of nucleation events is, however, subjective and overlapping or confusion within the classes may easily occur. To minimize the uncertainty of the classification method, Class 1 and Class 2 events are typically referred to as clear or intensive nucleation events (Clear Event class), where all classification stages were clearly fulfilled, whilst Class 3 events are referred to as Weak Events. Baranizadeh et al. (2014) named Clear Event and Weak Event days as "Quantifiable" and "Non-
25 Quantifiable" days, respectively, which describes better the difference of those classes. Both formation and growth rates can be determined for Quantifiable days but not both for Non-Quantifiable days.

Other classes of days are:

- **Non-Event days (NE)** (Class_NE): Days with no NPF in the nucleation mode particle size range are classified into
30 Non-Event days. These days are also interesting because, e.g., differences in conditions (meteorology, gas concentration, precipitation) during event and non-event periods are often studied in order to get better understanding of processes behind NPF.

- **Class 0** (Class_00): Some of days do not fulfil the criteria to be classified either event or non-event day and they are classified as Class 0. In that class, it is difficult to determine whether a nucleation event has actually occurred or not.

- **Bad Data (BD)** (Class_BD): Days with some malfunction in the measurement system (e.g. too high or low particle concentrations, missing data in part of the day) are classified as Bad Data. In the final data analysis, those days are typically ignored.

## 2.3 Event classification with a deep convolutional neural network (CNN)

In this study, we developed a novel method to analyze NPF events automatically. The schematic of the approach is shown in Fig. 2 and described in detail in the Appendix A. We used a large, deep convolutional neural network (CNN) named AlexNet (Krizhevsky et al., 2012, 2017), which has originally been trained on millions of images, as a subset of the ImageNet database (Deng et al., 2009), to classify images into 1000 different categories. The AlexNet was a breakthrough method in image analysis when it was introduced in 2012 and thus it is widely used in image recognition applications (LeCun et al., 2015).

Since the model has originally been trained to recognize images of very common objects, e.g., keyboards, mice, pencils, cars and many animals, it has to be fine-tuned to recognize other images.

The AlexNet itself cannot recognize NPF events so we used a transfer-learning technique for fine-tuning the model for PSD images (daily contour plots of time series of aerosol size distributions). In the fine-tuning of CNN by transfer-learning, the

new features are quickly learned by modifying a few of the last layers using a much smaller number of images than in the training of the original CNN (Weiss et al., 2016; Shin et al., 2016; Yosinski et al., 2014; Pan and Yang, 2010). This strategy is very efficient for images because of the structure of the CNN; the first layers process the image by extracting some features so that it becomes more abstract and the lasts layers attribute the probabilities of the classes. In our setup, we modified only a few of the last layers because of the low number of available training data.

In a transfer learning process of our data set, we categorized PSD images into six different classes based on the manual event classification (Class 1, Class 2, Class 3, NE, Class 0 and BD, see Fig. 1) and used a subset of images for training and the rest of the images for evaluation of the training (testing). Three different fractions of images were used in the training stage (80 %, 50 % and 20%, see Table 1) in order to study the effect of image set size on the NPF event classification. Images for training

and testing were selected randomly (certain percent of each category) and this was repeated ten times to evaluate the statistical accuracy of the training and data classification. The transfer learning process and data analysis were computed using a Matlab program (version R2016b) with the support of the package of Neural Network Toolbox Model for AlexNet Network (version 17.2.0.0) using a Linux server (CPU: 2x Intel Xeon E5-2630 v3, 2.40 GHz,16 cores; RAM 264.0 GB; GPU: Nvidia Tesla K40c, CUDA ver. 3.5, 12 GB). We used the standard procedures (trainNetwork, classify) and options (solver: sgdm, initial

learn rate: 0.001, max Epochs: 20, mini-batch size: 64) as described in an example for deep learning by Matlab (MathWorks, 2017). In the transfer learning of Matlab AlexNet, we changed two layers of the net to fit our data: the number of recognized classes was reduced to six in the last fully connected layer (fc8) and the category's names in the classification output layer (output) were charged to the names of our event classes. One training session lasted about from a half to one hour with the

used server and programs. The AlexNet CNN has been originally implemented in CUDA, but it is also available in Matlab. Other programing languages provide also good libraries for implementing the AlexNet such as Python or C++, and dedicated frameworks such as TensoFlow, Theano or Caffe are good candidates for an easy implementation of the NN and the learning stage.

To find out the best performance for the NPF event classification, we tested three different sets of particle size distribution images (Image Set, jpg file format with bit depth of 24 bits) with different plotting areas of the daily-measured size distributions:

**Image Set 1.** Original image (see Fig. 1) without the title, axis labels and numbers (i.e. axes of the plot and data inside them), size 1801 x 951 pixels (file size ca. 180 kB).

**Image Set 2.** Only colored, measured part of original images (i.e. time 00:00 – 24:00, particle diameter 3 – 630 nm), size 1646 x 751 pixels (file size ca. 120 kB).

**Image Set 3.** As previous but only an active time for NPF events is considered (i.e. 06:00 – 18:00, diameter 3 – 630 nm), size 831 x 746 pixels (file size ca. 60 kB).

Image sets with the different size were tested because all images have to resize to 227 x 227 pixels, which is an input image size of the AlexNet. Resizing of images pixels was conducted by a Matlab imresize -function using default bicubic interpolation (MathWorks, 2018). Resizing of images can lost some information of measured PSDs needed in the NPF identification.

After transfer learning each dataset (three image sets with three different training-testing fractions of images, repeated 10 times), the accuracy and training success were evaluated. We calculated the average success rate (accuracy) of the transfer learned CNNs by comparing CNN-based and human-made classifications of the test images. We also combined some classes together in the result analysis since overlapping of the classes could have easily occurred in the classification, e.g., Class 1 and 2 were combined to a Clear Event class (Cl_1-2).

**2.4 Statistical methods**

The performance of the classification between different image sets and training rates in different event classes was compared with the Kruskal-Wallis test and with multivariate analysis of variance (ANOVA). The ANOVA was conducted with a robust fit function (rlm; Venables and Ripley, 2002; Huber, 1981) because the normality and homoscedasticity assumptions of ordinary least squares method were not completely fulfilled. All statistical analyses were performed in R-software (R Core Team, 2017).

**3 Results and discussion**

The success of the transfer learning process of CNN was evaluated by confusion matrices, which showed a classification accuracy of the CNN method over visual inspection, i.e., how a certain class classified by human (visually) were categorized in the CNN classification. An example of the confusion matrices for one training-testing run (Image Set 1, training-testing ratio of 50 %/50 %) is presented in Fig. 3. When the training data set was analyzed with the trained CNN (same data), the overall classification accuracy (i.e. a fraction of days with an equal classification) was ca. 98 % and accuracies at certain event classes varied between 94-100 % (see Fig. 3a). For all cases (all image sets and training ratios), mean accuracies of ten different runs varied from 93 % to 98 %. The results show that the method can easily classified training data sets with very high accuracy, indicating that the training process of CNN was very successful.

Table 2 shows a summary of the classification accuracy for all cases when the method was applied for testing data sets (different days). The overall accuracy (all classes) is about 63 % for all studied cases. If we consider individual classes in more detail, the highest accuracies are in classes of NE and BD (ca. 80 %) whereas the lowest accuracies are in Classes 2, 3 and 0 (ca. 45 %), followed by Class 1 (ca. 53 %). The highest accuracies in NE and BD classes are apparently due to easier classification compared to other classes, e.g., no particles at low particles size ranges, no intensive particle growth or absence of particle at all and unusual high particles concentration in a part of data (see Fig. 1). The classification of other days is more challenging because differences between classes are not so distinct and the choice between classes can be difficult. When the clear event classes (Class 1 and 2) are combined into one category (Clear Event class, Cl_01-02), the classification accuracy increased to ca. 75 %. Overall, a classification accuracy for Clear Events and Non-Event categories is ca. 75-80 %, which can be considered a very good first result for automatic NPF event classification. In general, the classification would be more consistent, by both human and CNN, if only two different classes are used for event days: Clear and Weak events or "Quantifiable" and "Non-Quantifiable" events as described by Baranizadeh et al. (2014). From "Quantifiable" days, it is possible to quantify basic parameters of NPF event, e.g., particle formation and growth rates. Thus, we recommend that NPF event days should be categorized into classes of "Quantifiable" (Q) and "Non-Quantifiable" (NQ) events, in addition to Non-Event (NE), Undefined (UD or Class 0) and Bad-Data (BD) classes in the future analysis. This would also increase the number of images of the event classes in the training of CNNs.

If we look at the results in more detail, we can see variation in results between different training fractions (number of days in training in different event classes are shown in Table 1). In the cases of the training fractions of 80 % and 50 %, the classification accuracy values are quite similar but the accuracy decreases when the training fraction is only 20 % (especially for clear event days). Statistical analysis between different training-testing ratios shows that the 50 % and 80 % training rates are equally precise in all comparisons. This means that we could get adequate classification performance already with 50% training fraction (i.e., 135-708 images per class). However, when the size of the training set was lowered to 20 % (54-283

images per class), the classification became more uncertain in many comparisons. At 20 %, statistically significant differences were found when all classes were analyzed together and with classes Cl-01-02, Class 00, Class 02 and Class BD, indicating that the number of training days was too low for precise classification (e.g. Class 1 had only 54 days for training). In summary, a training fraction of 50 % (minimum 135 images per class) is a good compromise between accuracy, reliability and number of training days for the used data set.

We also studied the effect of image size (Image Sets 1-3) for classification accuracy. Only when all event classes were inspected together, there was a statistically significant difference between the image sets. Image Set 3 had a slightly lower performance rate than the two other sets. When limiting the classes to smaller sub groups the differences were not statistically significant anymore. The result indicates that image sets including all daily-measured size distribution data (Image Sets 1 and 2) are more suitable for CNN analysis than Image Set 3 with a reduced analysis period (6:00-18:00). Although images have to resize to the fixed input size (227 x 227 pixels) for CNN analysis, there is no need to reduce the analyzed period to cover only the active time for NPF. In fact, it is better to use all daily measured data in the analysis although resizing of images might lose a part of the information.

As described earlier, a choice between different event classes is somewhat arbitrary and not an easy task even for trained researchers and thus overlapping between different classes may occur. To analyze this overlapping in more detail, we plotted how a manually classified class is distributed into different classes by CNN classification. Figure 4 shows an example of CNN classification distributions for Image Set 1 with training-testing ratio of 80 %/20 %. Similar classification distribution into different classes can also be observed from the confusion matrix for Image Set 1 with training-testing ratio of 50 %/50 % (only one computing run) in Fig. 3b. The results show that an overlapping occurs typically between the adjacent or similar type of classes. For instance, a manually classified Class 1 is categorized mainly into Classes 1 and 2, Class 2 into Classes 1, 2 and 3, etc. The minimum overlapping is for classes NE and BD, which are the easiest classes to categorize by researchers. Similar overlapping is also observed in other analyzed cases (image sets and training-testing ratios).

To study overlapping or misclassification between different classes, we look in more detail at cases when Clear Event days (Classes 1 and 2) by human-made classification are categorized to Non-Event days by CNN-based classification and vice versa. Figure 5 shows examples of those days; the left-hand plots are categorized to Clear events by human and right-hand plots by CNN, the first row is examples of human made misclassifications, the second row CNN misclassifications and the third row difficult situations for classification. For instance, Fig. 5f shows a case when the initial stage of NPF has not been observed (probably due to change of wind or a mixing of boundary layer) but clear growth of particles is observed later. In that case, CNN-based classification does not "recognize" the missing of initial particle formation in the smallest particle sizes (ca. 3-4 nm) and therefore the day is classified as a Clear event day (Class 1 and 2). An opposite situation is shown in Fig. 5e where a Class 1 day (classified by human) was categorized two times into Non-Event and once into Class 2 day by CNN in

different computing runs. In that day, NPF is clear but the concentration of formed particles is lower than for typical event days and this probably affects the accuracy of the CNN-based classification. A similar misclassification can be seen in Fig 5c. In contrast, Figures 5a and 5b show examples of misclassifications, which are due to clear human errors, probably the researcher has just written down a wrong event class number. A general overview is that clear misclassifications seem to occur more commonly in human-made analysis than CNN-based categorization, which indicates that the developed CNN-based method has a better reliability and repeatability than manual human-made classification.

Only a few reports on automatic data analysis of NPF events have been published so far. In very recent conference proceedings, Zaidan et al. (2017) introduced a machine learning method based on a neural network. Their method utilized particle size distribution data pre-processing (fitting of lognormal distributions to the data), feature calculations and extractions (e.g. mean size, standard deviation, etc.) and principal component analysis (PCA) before the final classification by the neural network. Their preliminary results show a classification accuracy of ca. 83 % for Hyytiälä (Finland) data in 1996-2014 when only event and non-event days were considered. When compared to our method, they used several pre-processing steps before classification by the neural network and they did not use pre-trained CNNs or image recognition in their classifier. Kulmala et al. (2012) introduced a procedure for automatic detection of regional new particle formation. The procedure is based on monitoring the evolution of particle size distributions and it includes several steps, e.g., data noise cleaning and smoothing, excluding data larger than 20 nm and calculating regions of high particle concentration in particle size distributions. In the final step, the method automatically recognizes event regions from size distributions (i.e. regions of interest, ROI) and determines, e.g., NFP event start times as well as particle formation and growth rates. This method is very straightforward and does not include any data analysis methods based on artificial intelligence. To our knowledge, this method has not been used routinely in NPF event analysis for large datasets. Junninen et al. (2007) used an automatic algorithm based on self-organizing maps (SOM) and a decision tree to classify NPF events in Hyytiälä for 11-year data. They taught five SOMs by tuning them for specific event types and used a decision tree to get the probability for the day to belong to three different event classes (Event, Non-event and Undefined). The overall accuracy of the method was ca. 80 %. When comparing to our method, they pre-processed the data by decreasing the size resolution to 15 bins with variable width size bins in log-scale (higher resolution in smaller particles) and time resolution to 24 steps (1 h average). Furthermore, they weighted small particles (diameter < 20 nm) by a factor of 10 in one of the SOMs. To our knowledge, this method has not been used routinely in NPF event analysis. Hyvönen et al. (2005) used data mining techniques to analyze aerosol formation in Hyytiälä, Finland and Mikkonen et al. applied similar methods to datasets recorded in SPC (2006) and in Melpitz and Hohenpeissenberg, Germany (2011). They studied different variables and parameters that may be behind NPF but they did not make automatic classification for NPF events. Furthermore, Vuollekoski et al. (2012) introduced an idea on an eigenvector-based approach for automatic NPF event classification but they did not report any results because the method was still under development and its performance was uncertain.

We have not yet tested the method at other sites. Basically, "banana type" events, non-event days and bad data should be recognized from other site data if figures are plotted roughly in a similar way (one-day plot, same size ranges and axes, and color map). The method analyses features from size distribution plots, which are quite similar in many cases in different sites. However, we suggest that the CNN should be transfer learned again for new sites in order to get best results, especially if the shapes of size distributions are different compared to those of training data, e.g., low tide events in coastal sites (O'Dowd et al., 2010; Vaattovaara et al., 2006) or rush hour episodes in urban environments (Jeong et al., 2004; Alam et al., 2003). We encourage researchers to use the method in other sites and report results in order to develop the method. The accuracy of classification could be improved, e.g., by tuning training parameters, optimizing a number of classes used in analysis (e.g. merging Classes 1 and 2) and using synthetic training data.

## 4 Conclusions

We have developed a novel method based on deep learning to analyze new particle formation (NPF) events. The method utilizes a commonly available pre-trained Convolutional Neural Network (AlexNet) that has been trained by transfer learning to classify particle size distribution images. To our knowledge, this is the first time when a deep leaning method, i.e. transfer learning of a deep Convolutional Neural Networks (CNN), has successfully been used to classify automatically NPF events into different classes, including several event and non-event classes, directly from particle size distribution images, as the researchers do in a typical manual classification. Although there are general guidelines for human-made NPF event classification, the classification is always subjective and, therefore, it can vary between researchers or even within one researcher. In many ambiguous cases, it is not easy to attribute an event to the "correct" event class, even for an experienced researcher. The quality of the classification can especially vary for long-term data sets, which have been analyzed by several researcher in different times. Furthermore, a wrong event class can be listed to database due to a human error, which reduces reliability of the classification. Therefore, an automatic method, which can manage the whole dataset at once with a high reproducibility, is desired for NPF event analysis.

Our results show that transfer learning of a pre-trained CNN to recognize images of particle size distributions is a very powerful tool to analyze NPF events. The event classification can be done directly from existing data (figures) without any pre-processing of the data. Although an average classification accuracy of certain classes is ca. 65 %, the overall accuracy is ca. 80 % for Non-Event (NE) and Clear Event classes (Classes 1 and 2 combined), which is a good first result for automatic NPF event analysis. The most of miss-classified days have been categorized into the adjacent classes, which can be ambiguous to distinguish from each other. A compassion between CNN-based and human-made classification also showed that often the difference in categorization is due to a wrong or an incorrectly listed classification by researcher. Human-made classification can easily vary by people to people and can change over time whereas CNN-based classification is consistent all times. The CNN based categorization seems, at least, to be as reliable as human-made and it could be even more reliable if training image

sets are selected carefully. Typically, an analysis of large size distribution dataset requires manual labor and training for several researchers, which is very time consuming and quality of analysis may vary. The developed automatic CNN-based NPF event analysis can be used to study long-term effects of climate change on NPF in more efficiently, accurately and consistently, which helps us to understand in detail aerosol-climate interactions.

The transfer learning of pre-trained CNNs (like Alexnet and GooLeNet) allows us to make automatic event classification systems effectively without long-lasting design, training and computing of CNNs. Typically, a training of a CNN needs from thousands up to a million images but in the transfer learning of a pre-trained CNN, a hundred images can be enough for a precise classification. The pre-trained CNNs, as well as other novel machine learning and artificial intelligence methods, and

the increased computing capacity due to Graphics Processing Units (GPU) enable us to analyze very complex and large datasets as is typically in atmospheric science.

Instead of the transfer learning of pre-trained CNN, several other artificial intelligence methods could also be utilized in NPF analysis. The Recurrent Neural Network (RNN) is a good candidate for the classification and predication of time series

(Pascanu et al., 2013), however, not viewed as surface plots but as a sequence of particle size distributions. This allows more variability in the NPF classes, e.g. a continuum of NPF event intensity, and determining time dependent processes like particle formation and growth. Furthermore, the underlying weights of NNs would give insights about the size distribution evolution and this could be potentially used as an evolution model by substituting or complementing the general dynamic equation (GDE). In addition, the use of unsupervised learning method can help us describing new NPF event classes or merge classes

based on humanly imperceptible features. Reinforcement learning (Kaelbling et al., 1996) is an interesting technique especially for the cases for which the classification is not well defined, e.g. if the classification varies from one human to another, because it only requires a rewarding system or policy. The rewards need to be positive if the prediction is satisfying and negative if not; hence, the reward values can be the difference between the number of person that classified in the predicted class and those who did not. However, it should be mentioned that this technique works best for prediction models, such as RNN.

Currently, the state-of-the-art in artificial intelligence are deep CNN based algorithms *AlphaGo Zero* and *AlphaZero* developed by DeepMind (Silver et al., 2017b; Silver et al., 2017a). Those algorithms achieved superhuman performance by *tabula rasa* reinforcement learning without human data or guidance and defeated human in the game of Go and the most dedicated program Stockfish in chess.

In summary, we encourage researchers to use the CNN-based NPF identification method to their own data because of the better reliability and repeatability compared to human-made classification. However, there are still some weaknesses that should be kept in mind, e.g., quality and quantity of data is crucial in the training process, supervised learning is needed, the method still needs quite a lot of computing power (GPU), the identification is not perfect and particle formation and growth rates cannot be determined using the current model. Furthermore, the CNN should likely be transfer learned again for new sites in order to

the get best results, especially if the size distribution evolution is very different compared to that of the site of the training data. We suggest that in training, ca. 150 days per class should be enough to get reasonable classification. Alternatively, simulated data could be used for training (Lähivaara et al., 2018) but we have not tested how well it works in practice. The method is, however, very easy to use and results are accurate and consistent, especially for long-term data, if the CNN has been trained

carefully with high quality and reasonable amount of data. Finally, experiences and data obtained from other sites can be used in the further development of the method, e.g. to find suitable learning parameters, more data for training and possibility for a cross-validation of the method.

In the near future, we will analyze long-term changes in NPF in San Pietro Capofiume and utilize the method for other field

stations (e.g. Puijo, Kuopio, Finland). We are including more parameters into automatized NPF analysis (e.g. particle formation and growth rates, event start and end times) and are developing methods to predict NPF events based on meteorological and other atmospheric data. In addition, the simulation based deep learning is a potential research topic in the future (Lähivaara et al., 2018).

## 5 Appendix A

*General description of neural networks and training process*

We used a deep feedforward Neural Network (NN) to analyze NPF events in this study. Figure 2 shows a schematic of the used method and detailed descriptions of the method can be found in the textbooks of the subject (e.g. Buduma and Locascio, 2017; Duda et al., 2012). In general, the NNs are built by stacking interconnected layers of atomic units, or neurons. In each

layer, each neuron computes one very simple operation – often named activation – which consists of computing the weighted sum of the input variables plus a threshold and then passing the results to a function – e.g. tanh -function or Rectified Linear Unit (ReLU, i.e. $\max(0, x)$ -function)(Buduma and Locascio, 2017). To put it simply, a neuron emits a signal if the excitation signal – the input variable – reaches a threshold otherwise it is inactive. A single neuron cannot do much on its own, merely a linear discrimination, however, a set of connected and trained neurons, as a NN, can mimic human cognition abilities – the

emergence property. The depth – or number of layers – and the topology of a NN will determine what it can be used for. For instance, the recurrent NNs (RNN) are well designed for speech, text and time series classification or prediction (Pascanu et al., 2013) and convolutional NNs (CNN) outperform any other architecture at classifying visual data, i.e. images (Krizhevsky et al., 2012).

Before it can start classifying, a NN – of any kind – must be trained for the specific task, e.g. to recognize a car or a keyboard in an image. While the topology is, for most cases, assigned a priori, the parameters – weights and thresholds – are to be learned. The learning stage is crucial because it determines the efficiency of the classifier and, therefore, the machine learning

community has dedicated much effort to provide solutions for the issue – e.g. the backpropagation algorithm (Rumelhart et al., 1986) – and is still focusing on it decades after it all started (LeCun et al., 2015; Schmidhuber, 2015). There are as many learning methods as there are NN topologies, however they can be sorted in a few categories. For instance, the learning method may require a labeled training set, falling into the supervised learning (SL) category (Duda et al., 2012; Amari, 1998), or it

may learn by itself without an already classified set, in which case the method is referred to as unsupervised learning (UL) (Le, 2013; Radford et al., 2016). Another relevant classification of the learning methods is the depth of the NN. Even though there is no common consensus to define the limit between shallow and deep NN (DNN), it is commonly accepted to refer to deep NN as NN having at least two layers and very deep those with more than ten layers (Schmidhuber, 2015).

The learning stage is one of the main bottlenecks in an artificial intelligence, especially for NN. For instance, thousands of images are typically needed for learning in image recognition applications. Therefore, instead of relearning the complete structure if a new class is introduced or merged with another one, a less expensive strategy has emerged, namely transfer learning; it consists in using part of the existing learned parameters and learning only a subset of the structure. The pre-training method consists in first training the network with an UL algorithm and then continue the training with SL (Bengio et al., 2006).

Another solution is reinforcement learning, which is a technique involving an agent that learns policies based on interaction with its environment using trial-and-error. For this method, the correct states are not known – therefore it is an UL – but a system of rewards gives hints whether the predictions are correct or not (Mnih et al., 2013; Sutton, 1988; Kaelbling et al., 1996; Schmidhuber, 2015). Finally, the transfer learning method, which we have used in this study, consists of learning the weights of an NN that contains a lot of classes (see "General" frame of Fig. 2b) and then using those weights either as a starting

point for other learning sets with less classes or using partially the weights – e.g. for the first layers of the NN – and train only one part in order to specialize the NN as it is depicted the frame "Specialization" of Fig. 2b. (Yosinski et al., 2014; Cireşan et al., 2012; Pan and Yang, 2010; Mesnil et al., 2012; Krizhevsky et al., 2012; Weiss et al., 2016). For all kinds of NNs, the learning process is prone to learning too much detail – missing the generality of a semantic class – related to the examples. This phenomenon is known as overfitting and several methods have been developed to overcome this such as dropout (i.e.

ignoring randomly some neurons by setting them to zero during training)(Srivastava et al., 2014; Hinton et al., 2012).

*Convolutional neural network AlexNet and its transfer learning*

In this study, we used a CNN because it mimics the visual cognition process of the human. Figure 2c illustrates the structure
of the whole CNN and Fig. 2d and 2e some layers of the CNN in detail. Instead of being fully connected, the CNN is only locally connected, in other words, a neuron of a layer is connected to a compact subset of neurons of the previous layer. For every subset, the neurons compute the same operation – the parameters are shared across the neurons of a layer – resulting in a convoluted version of the input signal; hence the name. It is a good model for human vision because it applies the convolution operation throughout the image field – thus it has drawn a lot of attention in the image processing community (e.g. LeCun et

al., 1998; Cireşan et al., 2011; Chellapilla et al., 2006). CNN became popular when the large, deep CNN named AlexNet outperformed by far all the other pattern recognition algorithms during the ImageNet Large Scale Visual Recognition Challenge, ILSVRC 2012 (Krizhevsky et al., 2012). Since then, CNN has been improved for each classification challenge, starting by the ZFnet (Zeiler and Fergus, 2014) then by the deeper GoogleNet (Szegedy et al., 2015) and finally by the deepest

Microsoft ResNet (He et al., 2016).

AlexNet was used to identify NPF events in this study. The architecture of the AlexNet, shown in Fig 2c, consists of five convolutional layers (CL, Fig. 2d), some of which are followed by max-pooling (MP) layers (a down sampling layer that extracts the maximum values of predefined subregion) and three fully connected layers (DNN, Fig. 2e) with a final 1000-way

softmax (an output layer), which assigns a probability ($P(\ell)$) to each of the classes' label ($\ell$) for the input image (using a normalized exponential function, softmax). The first five layers of the AlexNet extract abstract features that are easier to classify than the original input image. This abstraction is computed by chaining up CL and MP layers. The CL (depicted in Fig. 2d) computes several convolution of the image by several kernels (i.e. matrices used in the convolution) – that must be learnt – and generates a multidimensional output – one dimension per kernel. For instance, in Fig. 2d the two kernels $v_1$ and

$v_2$ are applied to two compact sub regions of the data (*field 1* and *field 2*) by two units (*unit 1* and *unit 2*), which generate two 2D outputs ($y_1 = (x_1^1, x_1^2)$ and $y_2 = (x_2^1, x_2^2)$). The CL is what makes the CNN similar to the human visualization system because it applies the same transformation throughout the field of view – making the classification shift invariant. The MP layer merely computes a smaller image from each output image of the CL, e.g., an image of 256 x 256 pixels generated by CL will be reduced to smaller image of 64 x 64 by calculating maximum values of the 4 x 4 patches of the original image. The

last layers of the AlexNet is a DNN as depicted in Fig. 2e , i.e. a NN whose input and output layers are connected by hidden layers of neurons. Contrary to the CL, each neuron of a DNN's layer is connected to all the neurons of the previous layer and the parameters (weights and threshold) are not shared across the layer; this is the most general setting for a feedforward NN. In the AlexNet, the output DNN is the decision making center, which would be the analog to the abstraction stage created by the brain as the output of the visual system of a human. To speed up the learning process and reduce overfitting in the fully

connected layers (DNN), batch normalization and dropout regularization are employed in the AlexNet as well as the activation function ReLU.

The transfer learning process of the AlexNet (see Fig. 2b) is a two-stage optimization method. At first, the network is trained for a learning dataset ($S_g$) composed of the data itself ($x_k$) and its labels ($d_k$). This is achieved by solving an optimization

problem for all the NN's parameters that are represented by the vector sets $V$ and $W$, the first layers and the last layers, respectively. The optimization problem is defined so that the cost function $\mathcal{E}$ reaches a minimum for some optimal sets of parameters $\hat{V}$ and $\hat{W}$. Once the optimal parameters are known, for the most of elements $x_k \in S_g$, the output of the NN – $J(x_k | \hat{V}, \hat{W})$ – predicts the actual label $d_k$, which means that the NN is ready to classify data of the same semantic field as this

of the learning set. The second stage of transfer learning consists of optimizing only a few layers amongst the last ones – or possibly all the layers – using as an initial guess the optimal sets of the general learning problem. For instance, the cost function $\mathcal{E}$ in Fig 2b may be the same for the general and specialized learning step, but for the first stage, all the parameters are allow to vary while only the $W$ may vary during the specialization. This strategy is very efficient for images because of the structure of the CNN; the first layers process the image by extracting some features so that it becomes more abstract and the last layers attribute the probabilities of belonging to the classes. In our setup, we modified only a few of the last layers because of the low number of available training data $S_s$, and used original weights and thresholds in the other layers.

*Competing interests*. The authors declare that they have no conflict of interest.

*Acknowledgements*. The study was financially supported by the strategic funding of the University of Eastern Finland and the Academy of Finland (decisions no. 307331 and no. 250215). SM acknowledges financial support from the Nessling Foundation. Leone Tarozzi (ISAC-CNR) is acknowledged by his technical support in field operations of the DMPS in SPC.

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

**Figures:**

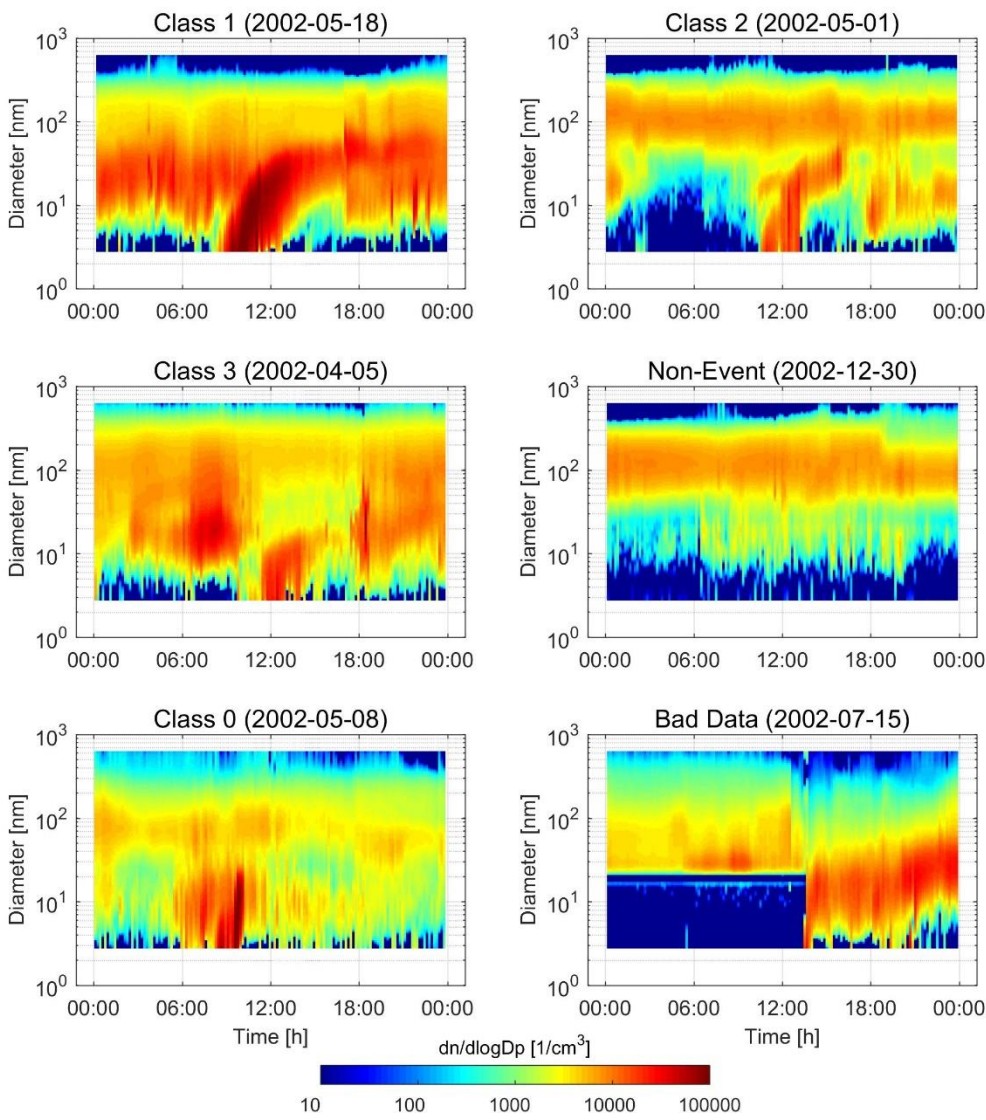

**Figure 1: Examples of particle size distributions (time in x-axis, particle diameter in y-axis and particle concentration presented by different colors) for different NPF event classes. Name of event class is indicated in plot title and date of measurement in brackets.**

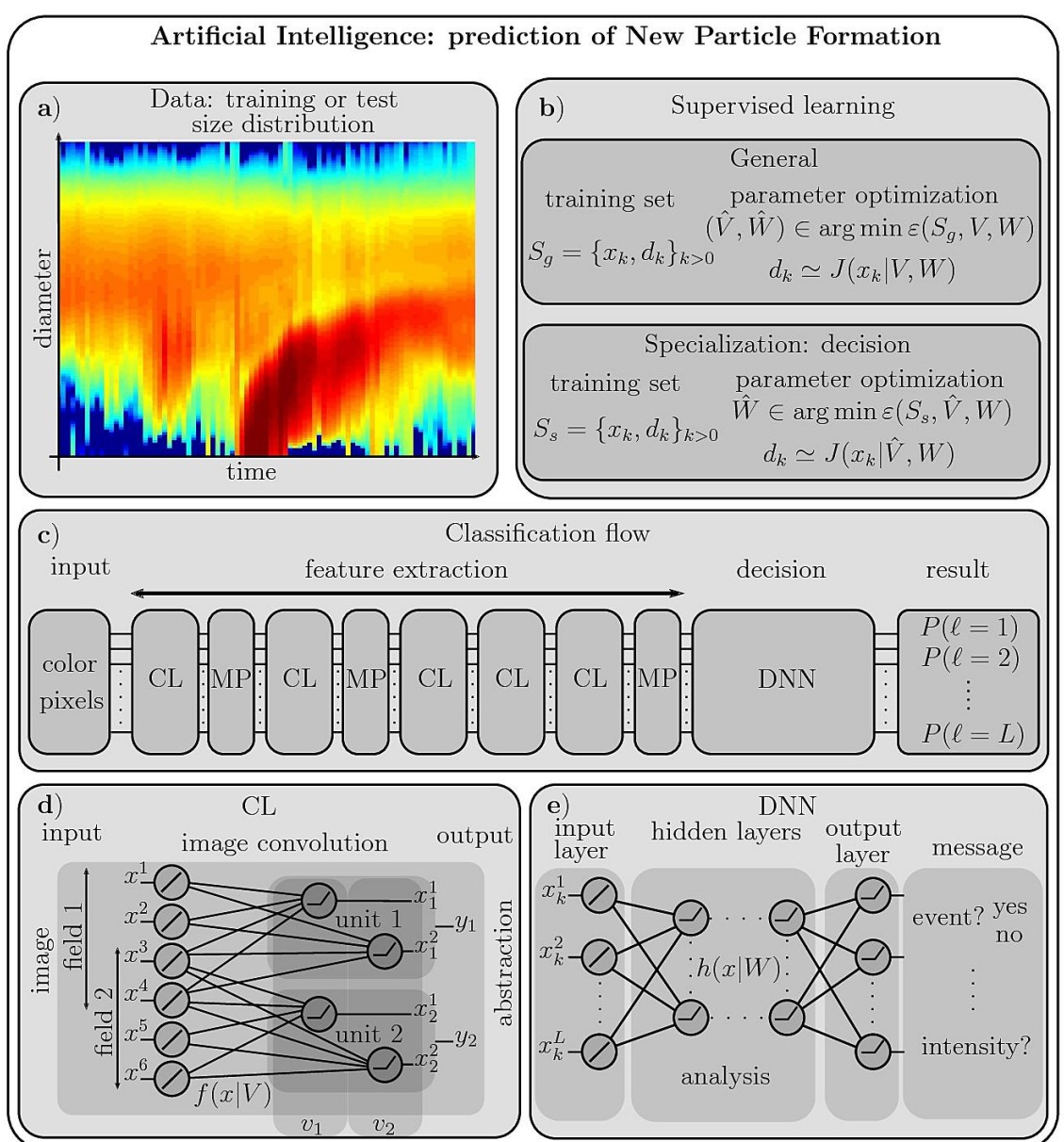

**Figure 2: Visual summary of the classification method: a) a typical dataset with a NPF event, b) the learning process of the optimization problem, c) the classification flow of the Convolutional Neural Network (CNN) and the two types of layers of the CNN, d) the Convolutional Layer (CL) and e) the Deep Neural Network (DNN), involved in the total CNN. The classification flow (c) is from the AlexNet CNN (Krizhevsky et al., 2012) composed of 5 CL intertwined with Max Pooling (MP) layers followed by a fully connected DNN of 3 layers. The method and variables in the figure are described in detail in Chapter 2.3 and Appendix A.**

**a)**

**Training data, Accuracy: 97.7%**

| Visual Method \ CNN Method | CI_01 | CI_02 | CI_03 | CI_NE | CI_00 | CI_BD |
|---|---|---|---|---|---|---|
| CI_01 | 100.0% 135 | 0.0% 0 | 0.0% 0 | 0.0% 0 | 0.0% 0 | 0.0% 0 |
| CI_02 | 8.1% 11 | 94.0% 203 | 0.3% 1 | 0.0% 0 | 0.0% 0 | 0.3% 1 |
| CI_03 | 1.5% 2 | 2.8% 6 | 95.2% 295 | 0.4% 3 | 1.0% 4 | 0.0% 0 |
| CI_NE | 0.0% 0 | 0.0% 0 | 0.0% 0 | 99.7% 706 | 0.5% 2 | 0.0% 0 |
| CI_00 | 0.0% 0 | 0.0% 0 | 0.0% 0 | 2.4% 17 | 96.0% 404 | 0.0% 0 |
| CI_BD | 0.0% 0 | 0.0% 0 | 0.0% 0 | 0.1% 1 | 0.0% 0 | 99.7% 300 |

**b)**

**Testing data, Accuracy: 63.8%**

| Visual Method \ CNN Method | CI_01 | CI_02 | CI_03 | CI_NE | CI_00 | CI_BD |
|---|---|---|---|---|---|---|
| CI_01 | 74.6% 100 | 14.0% 30 | 0.6% 2 | 0.0% 0 | 0.2% 1 | 0.3% 1 |
| CI_02 | 59.7% 80 | 47.4% 102 | 7.4% 23 | 0.1% 1 | 1.7% 7 | 0.7% 2 |
| CI_03 | 12.7% 17 | 41.4% 89 | 29.1% 90 | 5.9% 42 | 16.0% 67 | 1.3% 4 |
| CI_NE | 0.0% 0 | 1.9% 4 | 2.6% 8 | 86.9% 615 | 17.9% 75 | 2.0% 6 |
| CI_00 | 2.2% 3 | 8.4% 18 | 9.7% 30 | 27.0% 191 | 41.0% 172 | 2.0% 6 |
| CI_BD | 2.2% 3 | 1.4% 3 | 2.3% 7 | 3.1% 22 | 3.3% 14 | 83.7% 251 |

**Figure 3: Confusion matrices of a) training and b) testing data sets from one run of Image Set 1 with training-testing ratio of 50 %/50 %. Numbers (percent and absolute number of days) in rows indicate how a certain class classified by human (visually) were categorized in the CNN classification.**

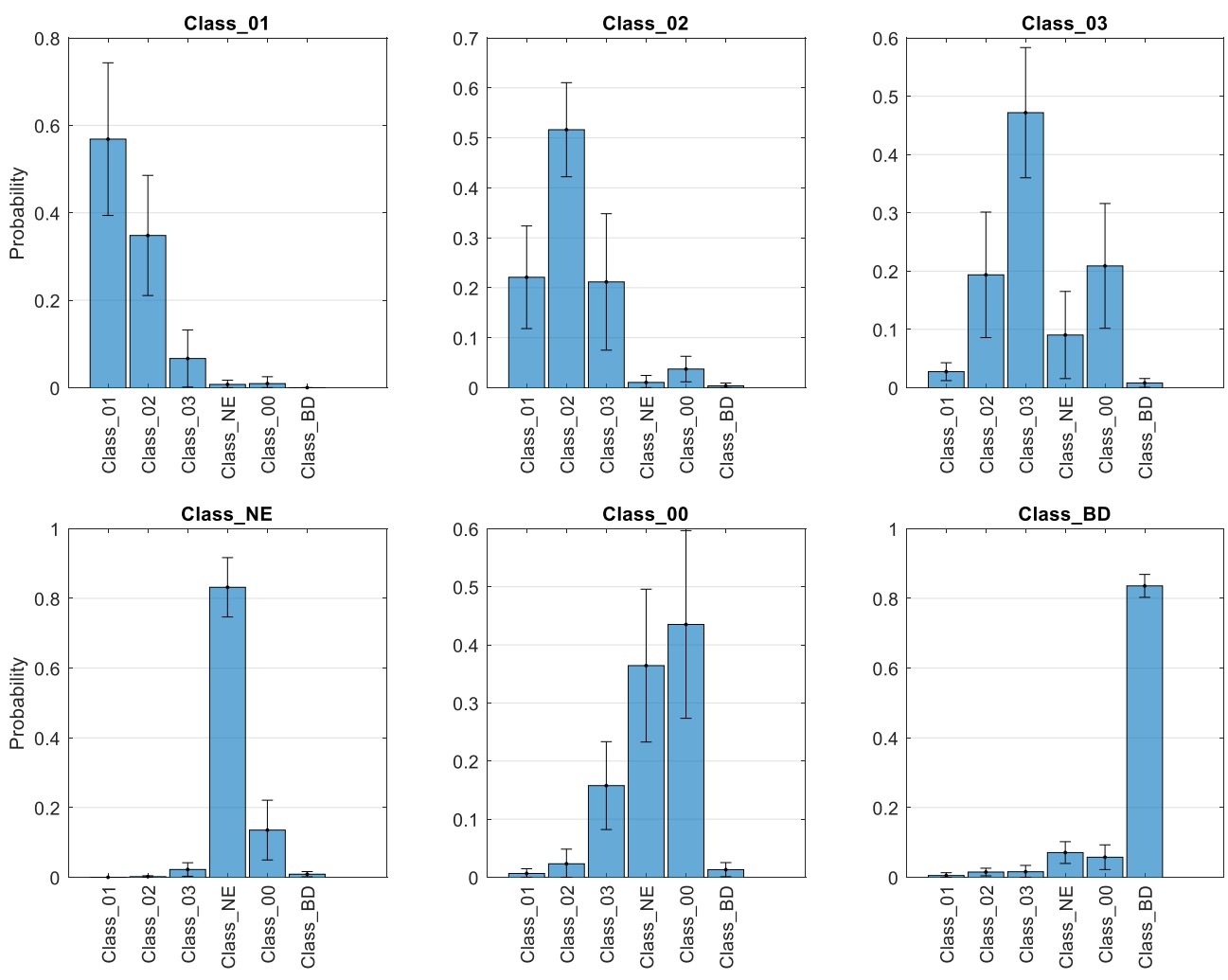

**Figure 4: Probability plots of distribution of CNN classification (x-axis labels) for different human-made classifications (indicated in a title of the plot), for Image Set 1 with training-testing ratio of 80 %/20 %. Histograms are mean values of ten different training-testing runs and error bars indicate standard deviations of the results.**

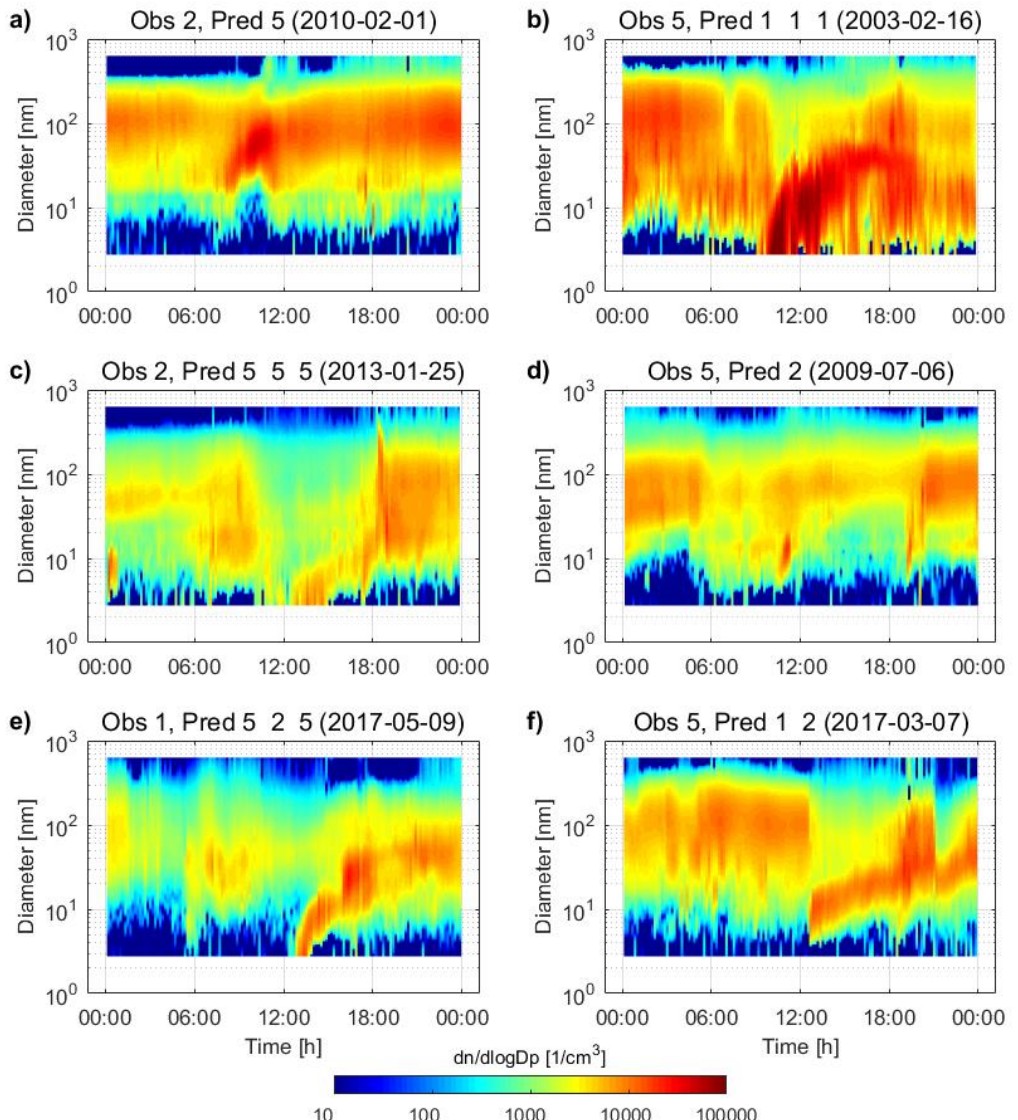

**Figure 5: Examples of particle size distributions for different days when visually categorized clear event days (Classes 1 and 2) are classified Non-Event days by CNN classification or vice versa. In title, numbers after *Obs* and *Pred* show NPF event classes determined with visual and CNN-based classification, respectively (date of measurement in brackets). Several event class numbers in Pred-cases are results from different calculation runs. The first row (a and b) contains examples of human made misclassifications, the second row (c and d) CNN misclassifications and the third row (e and f) ambiguous situations (low particle concentrations and change in air masses during day, respectively). CNN classification is from Image Set 1 with training-testing ratio of 80 %/20 %.**

**Tables:**

Table 1. Summary of number of days in different NPF event classes based on manual classification and division of days to three different training (used in CNN learning) and testing (used in testing success of training) ratio categories (80 %/20 %, 50 %/50% and 20 %/80 %).

| Event class | Days Total | Days % | Training 80 % | Testing 20 % | Training 50 % | Testing 50 % | Training 20 % | Testing 80 % |
|---|---|---|---|---|---|---|---|---|
| Class_01 | 269 | 6 % | 215 | 54 | 135 | 134 | 54 | 215 |
| Class_02 | 431 | 10 % | 345 | 86 | 216 | 215 | 86 | 345 |
| Class_03 | 619 | 15 % | 495 | 124 | 310 | 309 | 124 | 495 |
| Class_NE | 1416 | 34 % | 1133 | 283 | 708 | 708 | 283 | 1133 |
| Class_00 | 841 | 20 % | 673 | 168 | 421 | 420 | 168 | 673 |
| Class_BD | 601 | 14 % | 481 | 120 | 301 | 300 | 120 | 481 |
| Total | 4177 | 100 % | 3342 | 835 | 2091 | 2086 | 835 | 3342 |

Table 2. Summary of classification accuracy (%) of the transfer learned CNN when applied for test datasets (mean value ± standard deviation from ten model simulations) for three image sets with three different percentage of training images. Some classes have been merged together: CL_01-02 is a combination of Class_01 and Class_02, etc.

| Event class | Image Set 1 80 % | Image Set 1 50 % | Image Set 1 20 % | Image Set 2 80 % | Image Set 2 50 % | Image Set 2 20 % | Image Set 3 80 % | Image Set 3 50 % | Image Set 3 20 % | Overall |
|---|---|---|---|---|---|---|---|---|---|---|
| All_Class | 65 ± 1 | 64 ± 1 | 61 ± 1 | 65 ± 1 | 64 ± 1 | 60 ± 3 | 62 ± 3 | 63 ± 1 | 60 ± 2 | 63 ± 2 |
| Class_01 | 57 ± 17 | 54 ± 20 | 51 ± 15 | 63 ± 11 | 38 ± 10 | 53 ± 27 | 54 ± 17 | 54 ± 10 | 47 ± 22 | 52 ± 7 |
| Class_02 | 52 ± 9 | 51 ± 15 | 31 ± 17 | 40 ± 17 | 57 ± 14 | 27 ± 19 | 47 ± 20 | 45 ± 14 | 45 ± 21 | 44 ± 10 |
| Class_03 | 47 ± 11 | 32 ± 11 | 50 ± 14 | 46 ± 12 | 42 ± 14 | 48 ± 14 | 40 ± 10 | 44 ± 10 | 40 ± 16 | 43 ± 6 |
| Class_NE | 83 ± 9 | 80 ± 8 | 81 ± 13 | 80 ± 10 | 80 ± 10 | 81 ± 18 | 77 ± 12 | 81 ± 10 | 79 ± 11 | 80 ± 2 |
| Class_00 | 44 ± 16 | 53 ± 13 | 38 ± 18 | 51 ± 13 | 48 ± 12 | 36 ± 28 | 49 ± 12 | 43 ± 14 | 41 ± 14 | 45 ± 6 |
| Class_BD | 84 ± 3 | 85 ± 3 | 83 ± 3 | 85 ± 4 | 86 ± 2 | 80 ± 2 | 83 ± 4 | 84 ± 2 | 80 ± 2 | 83 ± 2 |
| Cl_01-02 | 81 ± 11 | 85 ± 9 | 62 ± 19 | 76 ± 11 | 78 ± 12 | 62 ± 19 | 81 ± 11 | 77 ± 13 | 71 ± 24 | 75 ± 8 |
| Cl_01-02-03 | 84 ± 6 | 79 ± 6 | 81 ± 6 | 82 ± 3 | 81 ± 5 | 80 ± 8 | 83 ± 6 | 82 ± 5 | 82 ± 3 | 81 ± 2 |
| Cl_03-NE | 77 ± 10 | 69 ± 10 | 78 ± 12 | 74 ± 8 | 73 ± 7 | 77 ± 18 | 70 ± 10 | 75 ± 10 | 72 ± 10 | 74 ± 3 |
| Cl_03-00 | 63 ± 10 | 60 ± 7 | 61 ± 13 | 66 ± 10 | 63 ± 13 | 60 ± 18 | 62 ± 10 | 60 ± 9 | 58 ± 14 | 61 ± 2 |
| Cl_NE-00 | 90 ± 5 | 93 ± 3 | 90 ± 4 | 92 ± 2 | 91 ± 3 | 91 ± 5 | 91 ± 3 | 90 ± 4 | 90 ± 2 | 91 ± 1 |
| Cl_03-NE-00 | 93 ± 4 | 91 ± 4 | 94 ± 4 | 93 ± 3 | 93 ± 3 | 95 ± 3 | 92 ± 4 | 93 ± 3 | 91 ± 6 | 93 ± 1 |