# Peer review of "Identification of new particle formation events with deep learning"

_Atmospheric Chemistry and Physics, 2017_

## Referee Comment (RC1) · Anonymous Referee #1 · 19 Feb 2018

**Summary**

This paper presents the use AlexNet, a specific class of deep learning method for automatize new particle formation identification. The study is interesting and important to substitute the current manual based NPF days classification to ease the study of atmospheric sciences.

[Figure]

**Response overview**

This article explains well the importance of applying data-driven method to replace human-based classification. In general, the materials, such as the used data, and the method are presented well. The obtained results are also adequate and interesting. The plots are clear and made well.

However, the reviewers would like to give some constructive comments, mentioning in the following paragraphs.

First, the article claimed that in the abstract and throughout the paper (section 1: lines 29-30) that this is the first time that their method classified successfully NPF events. Of course this statement is not true. In fact, Junninen et al. (2007), Kulmala et al. (2012) and later Zaidan et al. (2017) also succeed classifying this automatically. Although they were using different methods and aiming slightly for different NPF classes, but the primary aim is the same: automatic NPF event classification. Zaidan et al. (2017) also obtained the accuracy about 84% in their recent study, which they may considered also a success. Therefore, the word "first time" here undermines the previous contributions. Instead, you should mention how different this method and/or how this method is better compared to the previous methods.
**References:**

- Junninen, Heikki, et al. "An Algorithm for Automatic Classification of Two-dimensional Aerosol Data." Nucleation and Atmospheric Aerosols. Springer, Dordrecht, 2007. 957-961.

- Kulmala, Markku, Tuukka Petäjä, Tuomo Nieminen, Mikko Sipilä, Hanna E Manninen, Katrianne Lehtipalo, Miikka Dal Maso, et al. 2012. "Measurement of the nucleation of atmospheric aerosol particles." Nature protocols 7 (9): 1651–1667.

- Zaidan, M.A, V. Haapasilta, R. Relan, H. Junninen, P.P. Aalto, F.F. Canova, L.

Laurson, and A.S. Foster. Neural network classifier on time series features for predicting atmospheric particle formation days. In The 20th International Conference on Nucleation and Atmospheric Aerosols, 2017

In the lines 21-22 (section 1), the article said that "both of these studies were able to construct models to predict the probability of NPF occurrence with reasonable accuracy." This statement is not quite true. In fact, the paper by Hyvonen et al (2005) used data mining, such as ML classifier, to find relevant atmospheric variables to NPF. It is not to construct models to predict the probability of NPF. This statement should be revised.
**Reference;**

- Hyvönen, S., Junninen, H., Laakso, L., Dal Maso, M., Grönholm, T., Bonn, B., Keronen, P., Aalto, P., Hiltunen, V., Pohja, T., Launiainen, S., Hari, P., Mannila, H., and Kulmala, M.: A look at aerosol formation using data mining techniques, Atmos. Chem. Phys., 5, 3345-3356, 10.5194/acp-5-3345-2005, 2005.

The introduction section seems redundant and too long. There is a mixed reviews between the important of aerosol study, the use of data driven method and the algorithm review that is used in atmospheric study. This section should be narrowed down, by focusing only motivation of aerosol study and the use of data mining algorithms, such as CNN, in the field of atmospheric sciences and related discipline. General algorithm review can be pointed out to a specific reference sources.

In section 2 , lines 7-9, is it possible to get the latest statistics? The figure is 11 years old, this may be interesting to present the latest one, because the analyzed dataset was started from March 2017 (section 2: line 23).

Also, I may misunderstood your statement in section 2, lines: 23-25. Could you please clarify how the total days was 5534 from 24 March 2012 until 16 May 2017? (since in a year, there is only 365/366 days).

The explanation of class categorizes are clear and help readers understand.

For section 2.3, the first paragraph seems containing a lot of new details about the properties of CNN. This needs more clarification, for example,what are kernels, RELU and sofmax functions. Please also include other relevant mathematical details in the section 2.3? For example, in addition to Figure 2, it is good to include the mathematical representation of CNN.

Any justification why you used standard procedure and options? (line 17, section 2.3)

Section 2.4, line 13, what happened to the first reference?

Another part of your data pre-processing is to uniform the pixel size of the images, could you explain how this has been done? Is it an automatic process?

What is the value to estimate class 0? because human might put some ambiguous days into this class., which they are not sure if the days are event and non-event. This class is very subjective, the CNN learning in this subjective class may confuse the model and bias the results.

What do you think if you filter out the bad data before you feed this into CNN learning and analysis? In this case, the CNN learning can be simplified by reducing the number of classes.

There is no method that is perfect. Describe the weakness of your method in the conclusion part?

---

## Referee Comment (RC2) · Anonymous Referee #3 · 7 May 2018

General comments

The approach to use a Neural Networks (NN) method to automatize classification of new particle formation events is very useful in aerosol research at hundreds of previous, current or future atmospheric sites, where particle number size distributions are measured. It facilitates objective classification and faster classification, where you have long data sets. Hence, based on the research needs, this article deserves publication.

I therefore recommend the paper to be accepted subject to only minor revisions.

The minor concerns that must be addressed are:

You need to explain what happens when you have a different site: do you need to remake the training and testing with the new subset of images? Or can you use the

developed recognition in this study without any changes? And what happens when you have a site that has completely different shape of the size distribution compared to San Petro Capofiume? Or size range of the size distribution? In other words, what are your instructions and recommendations of how to proceed with your results and your methods when you do the same analysis at a different site? This needs to be clearly explained in abstract and conclusion sections. And there is no mentioning if you really recommend the method to be used already, or if you would like to develop it further before anyone should use it.

In relation to this: If you have to remake the training at each measurement site, do you always need to train the dataset with 50 % of the pictures? Does this mean that you have to select 50 % of your data at a new site already classified manually to be able to do the classification with NN? If this is the case, it is a severe drawback of the method. If you have to classify 50 % of the days manually on each new site, then then there is little point of doing the NN classification. If this is the case, then please write it clearly in the abstract and conclusions.

Even if these severe drawbacks occur, I still think the paper deserves publication, since a disadvantegeous result is important to convey to the science audience wanting to automatize classification of new particle formation events.

Some misspellings and grammatical errors. Please correct accordingly.

Thereafter, I go through some other minor issues below:

Abstract

The abstract is unusually long and has a very long description of the deep learning process. However, this is justified in this case, since aerosol researchers are normally not working with deep learning and a longer description is useful. So, the abstract should not be shortened.

Introduction and chapter 2.3

To be able to understand the NN methods, one way is to either, be very theoretically minded with an ability to understand abstract concepts and base your knowledge on this paper and other articles describing the methods, or you have to be practically oriented and learn by doing and be shown practical examples. As an extremely practically oriented person without an abstract mind set, I have no chance of understanding the methods based on reading. However, this does not automatically disqualify your text. After all, the abstract thinking might understand it. Hence, we have to accept that some people will understand the text, and some will not. Those that will not understand, will have to be learned by extensive simple examples, or by a teacher with a few practical examples, or at specialized workshops, and maybe with support from pedagogical video clips. Since it is not your task to develop extensive pedagogical descriptive examples (which is beyond the page limit of normal scientific papers), we have to accept this pedagogical problem and leave it as it is without further changes. Introduction

Page 2, row 12: Please add that passing on the method of classifying new particle formation events to a second person(s) might lead to systematic bias. If the second person passes on the knowledge to a third person(s), the systematic bias could increase further, and so on. I have experienced this problem previously, and it is a serious problem with the manual classification, and gives further motivation to develop automatic methods.

Page 2, rows 13-15: Wrong referencing to effects. That aerosols affect radiative balance does not automatically mean that they influence the climate via the direct and indirect effect. Please rephrase into something like this: "...radiative balance of the Earth and therefore the climate. They affect the climate directly by either scattering incoming solar radiation back to space or by absorbing it. Indirectly, aerosols affect the climate via their role in cloud formation as cloud condensation nuclei (CCN)."

Page 2, rows 18-19: Please add to the text that also the direct effect is leading to a cooling.

Page 2, rows 19: Please avoid using everyday language like "some". Please write "part" instead.

Page 4, rows 30-31: "Therefore, the idea of reusing what is already known instead of re-learning from scratch every time a new class has come up.". This sentence is not grammatically correct. Materials and Methods

Page 6, row 26: Please write that it is the "traditional method" in the title. Otherwise it can be confused with your new method.

Page 9, row 28 until Page 10, row 2. You mention that you "also tested three different sets of particle size distribution images". The reader might understand that these 3 methods are additional ones to the original method, while I think you mean that these are all the three methods that you have. Please rephrase to make it clearer, maybe by avoiding using "also" in the first sentence.

Page 9, rows 12-13 and Page 10, row 6. Please refer to Table 1 when mentioning the training/testing procedure.

Results and discussion

Page 12, rows 24-25. Do you have statistics to support your claim?

Tables

Table 1 caption text does not make sense when reading for the first time without studying the manuscript in detail. Please explain shortly in the caption text what you mean with training and testing. I am aware that this is explained later in the text (chapter 2.3), but needs a short explanation also when you mention Table 1 for the first time. Alternatively, you can write in the Table caption that this training and testing is explained in chapter 2.3.

---

## Author Comment (AC1) · 18 Jun 2018

We thank the referee for valuable comments and suggestions to improve the manuscript (MS). We have considered the comments and will modify MS accordingly. Our detailed responses to the referee's comments are below.

Referee's comment 1:

First, the article claimed that in the abstract and throughout the paper (section 1: lines 29-30) that this is the first time that their method classified successfully NPF events. Of course this statement is not true. In fact, Junninen et al. (2007), Kulmala et al. (2012) and later Zaidan et al. (2017) also succeed classifying this automatically. Although they were using different methods and aiming slightly for different NPF classes, but the

primary aim is the same: automatic NPF event classification. Zaidan et al. (2017) also obtained the accuracy about 84% in their recent study, which they may considered also a success. Therefore, the word "first time" here undermines the previous contributions. Instead, you should mention how different this method and/or how this method is better compared to the previous methods.

Authors' response:

We thank the referee to point out the valuable work of other researcher and the conference paper by Junninen et al., 2007, which we have not noticed before. In the revised version of MS, we will point out that this is the first time when a deep learning method, i.e. transfer learning of a deep neural network, has been successfully applied in NPF identification using the unprocessed data. In addition, we will compare in more detail our method with previous studies (Junninen et al., 2007; Kulmala et al., 2012; Zaidan et al., 2017). We would also note that the "protocol" for event analysis described by Kulmala et al. (2012) does not include the classification method by Junninen et al. (2007), even if Junninen is a co-author in the protocol paper.

Referee's comment 2:

In the lines 21-22 (section 1), the article said that "both of these studies were able to construct models to predict the probability of NPF occurrence with reasonable accuracy". This statement is not quite true. In fact, the paper by Hyvonen et al (2005) used data mining, such as ML classifier, to find relevant atmospheric variables to NPF. It is not to construct models to predict the probability of NPF. This statement should be revised.

Authors' response:

Hyvönen et al. states already in their Abstract "Using these two variables it was possible to derive a nucleation probability function." Therefore, even though the main purpose of the paper was to find variables related to NPF they still reported the probability

function and thus the sentence in referred lines is correct. However, we will rephrase the sentence in the revised MS as follows: "Both of these studies were able to find the characteristic conditions for NPF event days in each site and it was seen that the conditions differ significantly. They were also able to construct models to predict the probability of NPF occurrence with reasonable accuracy, and this approach has also been used in the day-to-day planning of a complex airborne measurement campaign (Nieminen et al., 2015)"

Referee's comment 3:

The introduction section seems redundant and too long. There is a mixed reviews between the important of aerosol study, the use of data driven method and the algorithm review that is used in atmospheric study. This section should be narrowed down, by focusing only motivation of aerosol study and the use of data mining algorithms, such as CNN, in the field of atmospheric sciences and related discipline. General algorithm review can be pointed out to a specific reference sources.

Authors' response:

We agree that the introduction section is quite long. However, be believe that a general review of algorithms suitable for dataset classification would be interesting for audience of the journal. We will make the introduction section more compact and move details of CNN to an appendix in the revised version of MS.

Referee's comment 4:

In section 2 , lines 7-9, is it possible to get the latest statistics? The figure is 11 years old, this may be interesting to present the latest one, because the analyzed dataset was started from March 2017 (section 2: line 23).

Authors' response:

We have revised the statistic by the latest results by Nieminen et al. (2018).

Referee's comment 5:

Also, I may misunderstood your statement in section 2, lines: 23-25. Could you please clarify how the total days was 5534 from 24 March 2012 until 16 May 2017? (since in a year, there is only 365/366 days).

Authors' response:

The correct starting time is 24 March 2002, the error will be corrected in the revised version of MS.

Referee's comment 6:

For section 2.3, the first paragraph seems containing a lot of new details about the properties of CNN. This needs more clarification, for example, what are kernels, RELU and sofmax functions. Please also include other relevant mathematical details in the section 2.3? For example, in addition to Figure 2, it is good to include the mathematical representation of CNN.

Authors' response:

We will move detailed descriptions about CNN and training processes to the appendix in the revised MS including their short definitions. However, the detailed mathematical representation of a CNN will be quite long and include many variables and thus, we think, will most likely confuse the readers. The readers can read detailed description from textbooks of the subject, referred in the new version of the MS (e.g. Buduma and Locascio, 2017; Duda et al., 2012, Ch. 6.6.).

Referee's comment 7:

Any justification why you used standard procedure and options? (line 17, section 2.3)

Authors' response:

Our focus was to study suitability of a CNN based method for the event identification

and we did not like to fine tune learning parameters. Therefore, we used procedures and options introduced in deep learning examples by Matlab because they worked well. In general, an optimization of the training parameters would be a very time consuming study. However, we will study an effect of training parameters in detail in the future.

Referee's comment 8:

Section 2.4, line 13, what happened to the first reference?

Authors' response:

We do not understand this comment. Both references (Venables and Ripley, 2002; Huber, 1981) are listed in the references. ANOVA is analysis of variance.

Referee's comment 9:

Another part of your data pre-processing is to uniform the pixel size of the images, could you explain how this has been done? Is it an automatic process?

Authors' response:

Resizing images to 227 x 227 pixels was done automatically by a standard imresize -function using default bicubic interpolation in the Matlab code (see https://se.mathworks.com/help/images/ref/imresize.html). This will be mentioned in the MS

Referee's comment 10:

What is the value to estimate class 0? because human might put some ambiguous days into this class., which they are not sure if the days are event and non-event. This class is very subjective, the CNN learning in this subjective class may confuse the model and bias the results.

Authors' response:

This is a very good question. In this study, we used the same classes than in the

Interactive
comment
manual classification in order to compare human-made and CNN based results. In fact, we made some tests by ignoring Class BD but not Class 0. Ignoring Class 0 in the CNN analysis and classifying days that are not clearly event or non-event days (or bad data) into Class 0, it would be a very good idea to improve the model. We will test this in the future studies.

Referee's comment 11:

What do you think if you filter out the bad data before you feed this into CNN learning and analysis? In this case, the CNN learning can be simplified by reducing the number of classes.

Authors' response:

We made some tests by ignoring Class BD class but this did not significantly change the results (accuracy was practically the same). We feel that Class BD should be included to the CNN analysis because it is one way to recognize when the instrument was not working properly and also the accuracy to find those days is high (ca. 85 %). In fact, BD class helps us to ignore automatically days with a system malfunction during data analysis. In general, reducing the number of classes is an interesting way of simplifying the learning process, but it means that we have less data in the training set, making the learning stage more prone to overfitting.

Referee's comment 12:

There is no method that is perfect. Describe the weakness of your method in the conclusion part?

Authors' response:

Of course, there are still some weaknesses in the developed CNN-based NPF identification method, e.g., quality and quantity of data is crucial in the training process, supervised learning is needed, the method needs some computing power (GPU), the identification is not perfect, particle formation and growth rates cannot be determined,

etc. We will mention those weaknesses of the method in more detail in the revised MS.

References:

Buduma, N., and Locascio, N.: Fundamentals of Deep Learning: Designing Next-Generation Machine Intelligence Algorithms, O'Reilly Media, Sebastopol, CA, USA, 298 pp., 2017.

Duda, R., Hart, P., and Stork, D.: Pattern classification, 2nd edition, John Wiley & Sons, Inc., 2012.

Hyvönen, S., Junninen, H., Laakso, L., Dal Maso, M., Grönholm, T., Bonn, B., Keronen, P., Aalto, P., Hiltunen, V., Pohja, T., Launiainen, S., Hari, P., Mannila, H., and Kulmala, M.: A look at aerosol formation using data mining techniques, Atmos. Chem. Phys., 5, 3345-3356, 10.5194/acp-5-3345-2005, 2005.

Junninen, Heikki, et al. "An Algorithm for Automatic Classification of Two dimensional Aerosol Data." Nucleation and Atmospheric Aerosols. Springer, Dordrecht, 2007. 957-961.

Kulmala, Markku, Tuukka Petäjä, Tuomo Nieminen, Mikko Sipilä, Hanna E Manninen, Katrianne Lehtipalo, Miikka Dal Maso, et al. 2012. "Measurement of the nucleation of atmospheric aerosol particles." Nature protocols 7 (9): 1651–1667.

Nieminen, T., Kerminen, V.-M., Petäjä, T., Aalto, P. P., Arshinov, M., Asmi, E., Baltensperger, U., Beddows, D. C. S., Beukes, J. P., Collins, D., Ding, A., Harrison, R. M., Henzing, B., Hooda, R., Hu, M., Hõrrak, U., Kivekäs, N., Komsaare, K., Krejci, R., Kristensson, A., Laakso, L., Laaksonen, A., Leaitch, W. R., Lihavainen, H., Mihalopoulos, N., Németh, Z., Nie, W., O'Dowd, C., Salma, I., Sellegri, K., Svenningsson, B., Swietlicki, E., Tunved, P., Ulevicius, V., Vakkari, V., Vana, M., Wiedensohler, A., Wu, Z., Virtanen, A., and Kulmala, M.: Global analysis of continental boundary layer new particle formation based on long-term measurements, Atmos. Chem. Phys. Discuss., https://doi.org/10.5194/acp-2018-304, in review, 2018.

Zaidan, M.A, V. Haapasilta, R. Relan, H. Junninen, P.P. Aalto, F.F. Canova, L. Laurson, and A.S. Foster. Neural network classifier on time series features for predicting atmospheric particle formation days. In The 20th International Conference on Nucleation and Atmospheric Aerosols, 2017
* * *

---

## Author Comment (AC2) · 18 Jun 2018

We thank the referee for valuable comments and suggestions to improve the manuscript (MS). We have considered the comments and will modify MS accordingly. Our detailed responses to the referee's comments are below.

Referee's comment 1:

You need to explain what happens when you have a different site: do you need to remake the training and testing with the new subset of images? Or can you use the developed recognition in this study without any changes? And what happens when you have a site that has completely different shape of the size distribution compared to San Petro Capofiume? Or size range of the size distribution? In other words, what are

your instructions and recommendations of how to proceed with your results and your methods when you do the same analysis at a different site? This needs to be clearly explained in abstract and conclusion sections. And there is no mentioning if you really recommend the method to be used already, or if you would like to develop it further before anyone should use it.

Authors' response:

We have not yet tested the method with other sites. Basically, "banana type" events, non-event days and bad data should be recognized from other site data if pictures are plotted roughly in a similar way (one-day plot, size ranges, axes and color map). The method analyses features from size distribution plots, which are quite similar in many cases in different sites. However, we still think that the CNN should be transfer learned again for new sites in order to get best results, especially if the shapes of size distributions are completely different (e.g. low tide events in Mace Head in Ireland or rush hour episodes in urban environments). In other words, once learned CNN can be used in other sites but more precise results will be got if CNN has been transfer learned with data from the same site. In general, we think that anyone can already use the method because the basic concept is efficient enough. However, classification accuracy can be improved by testing different parameters and optimizing set of classes.

We will discuss this in more detail in the abstract and conclusions sections in the revised version of MS.

Referee's comment 2:

In relation to this: If you have to remake the training at each measurement site, do you always need to train the dataset with 50 % of the pictures? Does this mean that you have to select 50 % of your data at a new site already classified manually to be able to do the classification with NN? If this is the case, it is a severe drawback of the method. If you have to classify 50 % of the days manually on each new site, then then there is little point of doing the NN classification. If this is the case, then please write it clearly

in the abstract and conclusions.

Authors' response:

In general, the percentage of labeled data is not the most important parameter, but the number of images in relation to the size of the CNN is the relevant one. For instance, if the CNN is small – small number of layers and small number of neuron per layer – then, the minimum amount of images in the training pool is small, however, if the CNN is large, then the training pool should be large too. Furthermore, the complexity of the problem to be solved affects the number of images needed.

As a rule of thumb, if the data are similar enough from one site to another – particle concentration, time scale, measurement device, etc. – and the data are depicted with the same color map and log scale, then the classifier can be readily used. The color map plays a role only if it is not an "optimal" map; for instance, if the colormap distorts the data – e.g. a small variation in number concentration creates a big difference in color – then choosing another colormap will modify the performance of the classifier.

In our case, we have totally ca. 2000 days for training in the 50 % case and, e.g., Class 1 has only ca. 130 days. If you merge some classes together (e.g. Classes 1 and 2) and you have well-classified data, less training data is needed. Generally, you need some data for training. Once you have trained CNN, you can used it for all new data from that site. Alternatively, simulated data could be used for training but we have not tested how well it works in practice.

We will discuss this in more detail in abstract and conclusions sections in the revised version of MS. In summary, we suggest that the CNN should likely be transfer-learned again for new sites to get the best results but in training, ca. 150 days per class should be enough to get a reasonable classification.

Referee's comment 3:

Abstract

The abstract is unusually long and has a very long description of the deep learning process. However, this is justified in this case, since aerosol researchers are normally not working with deep learning and a longer description is useful. So, the abstract should not be shortened.

Authors' response: We will only slightly modify abstract based on the Referee #1 comments.

Introduction and chapter 2.3

Referee's comment 4:

To be able to understand the NN methods, one way is to either, be very theoretically minded with an ability to understand abstract concepts and base your knowledge on this paper and other articles describing the methods, or you have to be practically oriented and learn by doing and be shown practical examples. As an extremely practically oriented person without an abstract mind set, I have no chance of understanding the methods based on reading. However, this does not automatically disqualify your text. After all, the abstract thinking might understand it. Hence, we have to accept that some people will understand the text, and some will not. Those that will not understand, will have to be learned by extensive simple examples, or by a teacher with a few practical examples, or at specialized workshops, and maybe with support from pedagogical video clips. Since it is not your task to develop extensive pedagogical descriptive examples (which is beyond the page limit of normal scientific papers), we have to accept this pedagogical problem and leave it as it is without further changes.

Authors' response:

We have recognized this problem how to describe the method in a simple enough way but simultaneously theoretically enough. Based on the Referee #1 comments, we will move some of the most theoretical parts of the method description to an appendix to make the text more concise. Very practical descriptions and some examples (videos and codes) can be found, e.g. from Mathworks (Matlab) web pages: https://se.mathworks.com/discovery/deep-learning.html. Furthermore, more detailed description can be found from textbooks of the subject (e.g. Buduma and Locascio, 2017; Duda et al., 2012, Ch. 6.2.), which we now cite in the text.

Introduction

Referee's comment 5:

Page 2, row 12: Please add that passing on the method of classifying new particle formation events to a second person(s) might lead to systematic bias. If the second person passes on the knowledge to a third person(s), the systematic bias could increase further, and so on. I have experienced this problem previously, and it is a serious problem with the manual classification, and gives further motivation to develop automatic methods.

Authors' response:

We will add to revised MS that passing on the manual classification method from researcher to researcher could lead an increasing systematic bias.

Referee's comment 6:

Page 2, rows 13-15: Wrong referencing to effects. That aerosols affect radiative balance does not automatically mean that they influence the climate via the direct and indirect effect. Please rephrase into something like this: ". . .radiative balance of the Earth and therefore the climate. They affect the climate directly by either scattering incoming solar radiation back to space or by absorbing it. Indirectly, aerosols affect the climate via their role in cloud formation as cloud condensation nuclei (CCN)."

Authors' response: We will change text accordingly.

Referee's comment 7:

Page 2, rows 18-19: Please add to the text that also the direct effect is leading to a

cooling.

Authors' response: We will change text as suggested.

Referee's comment 8:

Page 2, rows 19: Please avoid using everyday language like "some". Please write "part" instead.

Authors' response: We will change text as suggested.

Referee's comment 9:

Page 4, rows 30-31: "Therefore, the idea of reusing what is already known instead of re-learning from scratch every time a new class has come up.". This sentence is not grammatically correct.

Authors' response: This sentence have been deleted when we have reorganized MS.

Materials and Methods

Referee's comment 10:

Page 6, row 26: Please write that it is the "traditional method" in the title. Otherwise it can be confused with your new method.

Authors' response: We will change the title as suggested.

Referee's comment 11:

Page 9, row 28 until Page 10, row 2. You mention that you "also tested three different sets of particle size distribution images". The reader might understand that these 3 methods are additional ones to the original method, while I think you mean that these are all the three methods that you have. Please rephrase to make it clearer, maybe by avoiding using "also" in the first sentence.

Authors' response: We will change text as suggested.

[Figure]

Referee's comment 12:

Page 9, rows 12-13 and Page 10, row 6. Please refer to Table 1 when mentioning the training/testing procedure.

Authors' response: We will change text accordingly.

Results and discussion

Referee's comment 13:

Page 12, rows 24-25. Do you have statistics to support your claim? Authors' response:

We have not studied this by statistical analysis and it is just a general statement based on randomly selected days. It would be very time-consuming if we check all misclassified days manually. In general, a part of misclassifications is pure booking errors made by researchers (wrong class written to database) and thus human-made errors seem to be more common. In addition, classification can be easily vary from researcher to researcher.

Tables

Referee's comment 13:

Table 1 caption text does not make sense when reading for the first time without studying the manuscript in detail. Please explain shortly in the caption text what you mean with training and testing. I am aware that this is explained later in the text (chapter 2.3), but needs a short explanation also when you mention Table 1 for the first time. Alternatively, you can write in the Table caption that this training and testing is explained in chapter 2.3.

Authors' response: We will change the caption text as suggested.